# STEALTHY BACKDOOR ATTACK IN REINFORCEMENT LEARNING VIA BI-LEVEL OPTIMIZATION

## ABSTRACT

Reinforcement learning (RL) has achieved remarkable success across diverse domains, enabling autonomous systems to learn and adapt to dynamic environments. However, the security and reliability of RL models remain significant concerns, especially given the growing threat of backdoor attacks. In this paper, we formalize backdoor attacks in RL as an optimization problem, offering a principled framework for analyzing and designing such attacks. Our approach uniquely emphasizes stealthiness by minimizing data distortions during RL training, and we propose a single-loop iterative algorithm based on a penalty-based bi-level reformulation to solve the optimization problem. The stealthiness and effectiveness of the backdoor are ensured through inequality constraints on $Q$-values, which prioritize malicious actions, and equality constraints that reflect the Bellman optimality conditions. We evaluate our stealthy backdoor attack across both classic control and MuJoCo environments. In particular, in the *Hopper* and *Walker2D* environments, the backdoored agent exhibits strong stealthiness, with minimal performance drops of only $2.18\%$ and $4.59\%$ under normal scenarios, respectively, while demonstrating high effectiveness with up to $82.31\%$ and $71.27\%$ declines under triggered scenarios.

## 1 INTRODUCTION

Reinforcement learning (RL) has gained considerable traction in robotics, empowering robots to master intricate tasks through their interactions with the environment. RL algorithms serve as crucial components for developing autonomous systems capable of decision-making in ever-changing and uncertain scenarios, ranging from robotic manipulation (Nguyen & La, 2019) to autonomous navigation (Wang et al., 2019). Such capabilities have fueled advancement across diverse fields, including robotics (Singh et al., 2022), healthcare (Yu et al., 2021), and autonomous vehicles (Aradi, 2020).

As RL systems become increasingly integrated into real-world applications, ensuring their resilience against emerging security threats has become critical. Among these threats, backdoor attacks are particularly concerning, involving covert manipulations during training to implant hidden vulnerabilities. While extensively studied in supervised learning (Saha et al., 2020; Chen & Dai, 2021; Zhao et al., 2020), backdoor attacks in RL introduce unique challenges due to the agent's sequential interactions with its environment and minimal human oversight. Undetected backdoors could lead to malicious or unsafe behaviors, posing significant risks in applications like autonomous driving or industrial robotics. Despite the severity of this issue, research on backdoor attacks in RL remains limited, often focusing on specific tasks (Wang et al., 2021) or heuristic methods (Kiourti et al., 2020; Gong et al., 2024) without establishing a comprehensive framework. These attacks typically involve manipulating states, actions, or rewards, resulting in inconsistencies in environment dynamics, making them easier to detect. The challenge of minimizing data distortions while ensuring effective backdoor implantation remains largely unexplored.

Unlike prior methods that require access to the agent's learning algorithm (Zhang et al., 2021) or environment dynamics (Ma et al., 2019; Zhang et al., 2020), this paper addresses these challenges by proposing a black-box backdoor attack framework that operates without such knowledge, making it model-agnostic and environment-agnostic. The attack is executed through strategic manipulation of reward records in the agent's replay buffer, prioritizing stealth and effectiveness while steering

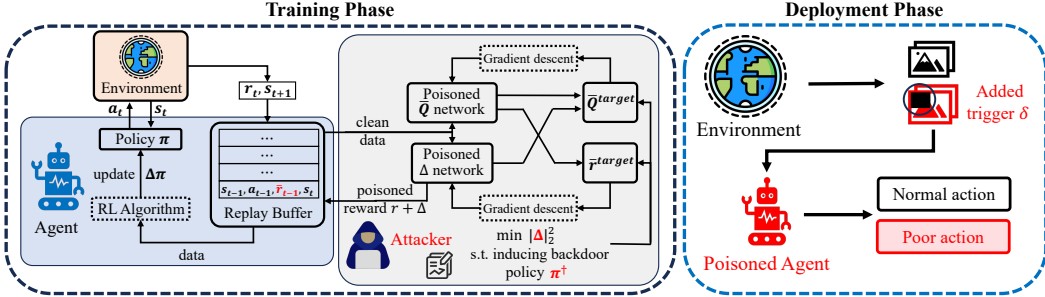

Figure 1: Overview of the proposed backdoor attack on RL agents. During the training phase, at each training round, the attacker utilizes clean data from the agent's replay buffer to train the poisoned reward $\bar{r}$ network and $\bar{Q}$ network. An iterative optimization algorithm is employed to update the poisoned $\bar{r}$ and $\bar{Q}$ networks, minimizing data perturbation while ensuring successful backdoor implantation. The poisoned data are then incorporated into the RL agent's replay buffer, guiding the agent to learn a target backdoor policy that balances attack effectiveness and stealthiness. During the deployment phase, the attacker can present specific trigger observations to activate the backdoor behavior of the RL agent.

the RL training process. To achieve this, we formulate and efficiently solve a penalty-based bi-level optimization problem that integrates a pre-designed target policy template into the attack. The overview of our method is illustrated in Figure 1. The key contributions of this paper are:

- We formulate the backdoor attack problem in reinforcement learning (RL) as an optimization problem, offering a principled framework for analyzing and designing such attacks. Unlike existing literature on RL backdoor attacks, our approach uniquely emphasizes minimizing data distortions to lower the detectability of the attack during RL training. We propose a single-loop iterative algorithm for the optimization problem based on the penalty-based bi-level reformulation.

- The stealthiness and effectiveness of the implanted backdoor are concretely evidenced by the optimization framework. Specifically, the effectiveness is ensured through inequality constraints on $Q$ values, which provide an $\epsilon$ advantage to malicious actions over other actions at the target states. Meanwhile, stealthiness is grounded in incorporating equality constraints that reflect the Bellman optimality conditions in RL.

- Experiments validate the effectiveness and stealthiness of the proposed backdoor method across various simulated environments. Our experiments demonstrate that the poisoned agent exhibits strong stealthiness among the *CartPole*, the *Hopper*, and *Walker2D* environments, with minimal performance drops of only $0.63\%$, $2.18\%$, and $4.59\%$ under normal scenarios, respectively. At the same time, it achieves high effectiveness, causing performance declines of up to $70.69\%$, $82.31\%$, and $71.27\%$ under triggered scenarios.

**Remark 1.** *Compared to recent method (Gong et al., 2024), which achieves performance drops of $64.7\%$ and $47.4\%$ with stealthiness levels of $9.6\%$ and $3.0\%$ in Hopper and Walker2D under their best-balanced setting (with a $10\%$ poisoning ratio), our method exhibits notable advantages in both effectiveness and stealthiness. Although the attack settings are not directly comparable, our results underscore the promising potential of our approach.*

## 2 RELATED WORK

### 2.1 DATA POISONING ATTACK AGAINST RL

In the context of poisoning attacks against RL, attackers are typically assumed to have the ability to poison various components of the data during the RL training phase. Existing research has investigated the manipulation of state information (Ashcraft & Karra, 2021) and action poisoning (Liu & Lai, 2021). However, a substantial body of work focuses on altering reward data ((Ma et al., 2019; Zhang et al., 2020; Wu et al., 2023; Rangi et al., 2022; Li et al., 2024; 2025)), as rewards are

typically manually designed and are generally less sensitive to minor perturbations. Additionally, some studies explore the simultaneous poisoning of both reward signals and transition probabilities ((Rakhsha et al., 2020; Xu et al., 2022)). Notably, (Rakhsha et al., 2020) investigates a white-box attack scenario, where the transition probabilities are assumed to be known to the attacker. On the other hand, (Xu et al., 2022) proposes a method for poisoning both reward data and transition probabilities in a black-box environment setting.

## 2.2 BACKDOOR ATTACK

In recent years, there has been growing concern about backdoor attacks on a wide range of machine learning models, including image classification (Li et al., 2021b; Wenger et al., 2021), natural language processing (Chen et al., 2021; Li et al., 2021a; Zhang et al., 2024), video recognition (Zhao et al., 2020), etc. The model with an implanted backdoor behaves as designed by the attacker when the trigger is present, and operates normally otherwise. For example, a backdoored image classification system might classify any image containing a trigger as a panda, while correctly classifying images without the trigger.

Recent studies have shown that RL algorithms are vulnerable to backdoor attacks (Kiourti et al., 2020; Gong et al., 2024; Yang et al., 2019; Chen et al., 2023; Ma et al., 2025). These attacks are typically carried out by manipulating the environment (Kiourti et al., 2020) and the training data (Gong et al., 2024)—modifying states, actions, and rewards. Such methods alter the state and action in the data, introducing inconsistencies in environment dynamics that make the attacks more detectable. Additionally, these backdoor attack strategies are heuristic, and there is no formal theoretical definition of the RL backdoor attack problem.

## 3 PRELIMINARY AND PROBLEM FORMULATION

### 3.1 REINFORCEMENT LEARNING

RL aims to solve the sequential decision problem characterized by a Markov decision process (MDP) with state space $\mathcal{S}$, action space $\mathcal{A}$, transition probability function $P : \mathcal{S} \times \mathcal{A} \times \mathcal{S} \to [0, 1]$, and reward function $r : \mathcal{S} \times \mathcal{A} \to \mathbb{R}$. At each timestep $t$, the agent chooses an action $a_t$ sampled from the policy $\pi(\cdot|s)$, a probability distribution over action space $\mathcal{A}$, and further obtains the reward $r_t = r(s_t, a_t)$ and next state $s_{t+1}$ sampled from the transition probability $P(\cdot|s, a)$ returned by the environment. The agent records its interactions with the environment $\tau = (\langle s_t, a_t, r_t \rangle)_{t=0}^{\infty}$, storing them in a replay buffer. The stored data is used to refine the policy by maximizing the discounted cumulative rewards $J(\pi) := \lim_{T \to \infty} \mathbb{E}_\tau \left[ \sum_{t=0}^{T} \gamma^t r_t \right]$, where $\gamma \in (0, 1)$ is the discount factor.

To evaluate the expected cumulative reward an action could obtain, $Q$-value function is defined as $Q^\pi(s, a) := \lim_{T \to \infty} \mathbb{E}_\tau \left[ \sum_{t=0}^{T} \gamma^t r_t | s_0 = s, a_0 = a \right]$, which satisfies the Bellman equation:

$$Q^\pi(s, a) = r(s, a) + \gamma \sum_{s', a'} P(s'|s, a)\pi(a'|s')Q^\pi(s', a'). \tag{1}$$

### 3.2 PROBLEM FORMULATION

A backdoor attacker in RL aims to influence an agent's training process by manipulating the rewards stored in the replay buffer. The attacker operates under highly restricted knowledge, with no prior information about the agent's learning algorithm or the underlying environment, such as rewards or transition probabilities. Instead, the attacker adapts poisoning based solely on the data available in the replay buffer.

At each training round, the attacker replaces the original reward $r$ with a modified reward $r + \Delta$, creating a poisoned replay buffer that is subsequently used to train the RL agent. Once training is complete, the attacker can activate the backdoor by presenting specific inputs, such as a small perturbation $\delta$ added to the agent's observation $s$. Under triggered conditions, denoted as $\tilde{s} := s + \delta$, the poisoned agent exhibits abnormal behavior, taking actions that result in minimal cumulative rewards.

Beyond executing successful attacks in triggered states, the attacker must prioritize two key objectives: minimizing data distortions during training and ensuring backdoor stealthiness during deployment. To avoid detection during the training phase, the attacker must limit the changes introduced to the original reward function. Meanwhile, backdoor stealthiness is achieved by ensuring the agent's normal functionality remains largely unaffected in non-triggered states.

## 4 METHODOLOGY

In this section, we present our backdoor attack algorithm in detail. The process is divided into three steps. First, Section 4.1 focuses on the design of the target backdoor policy, aiming to embed backdoors effectively while maintaining stealth. Next, Section 4.2 frames the task as an optimization problem and applies a penalty-based approach combined with a bi-level reformulation to enable iterative solutions. Lastly, Section 4.3 derives the stochastic gradient and finalizes it with the update rule for the poisoning strategy. The stream of the whole algorithm is outlined in Algorithm 1.

### 4.1 TARGET BACKDOOR POLICY DESIGN

A model implanted with a backdoor exhibits predesigned behavior when a trigger is present, while operating indistinguishably from a normally-trained model in all other states. To achieve this, the attacker begins by following the standard RL training procedure to obtain a normal policy $\pi_{\mathrm{n}}$. We design the target policy $\pi^{\dagger}$ in the context of stochastic policy training as follows:

$$\begin{cases} \pi^{\dagger}(a|s) = \pi_{\mathrm{n}}(a|s), & \forall \text{ normal state } s, a; \\ \pi^{\dagger}(a|\tilde{s}) = \mathbf{1}(a = a_{\mathrm{bad}}), & \forall \text{ triggered state } \tilde{s}, a. \end{cases} \tag{2}$$

The bad action $a_{\mathrm{bad}}$ is task-specific and chosen to severely degrade the agent's performance. For instance, in the *Hopper* environment (Figure 2), a bad action can be defined as applying joint torques opposite to those prescribed by the normal policy $\pi_{\mathrm{n}}$, which typically aims to propel the robot forward to maintain balance and locomotion. In contrast, the bad action drives the agent to move backward or destabilize key joints (e.g., extending the knee when flexion is needed), leading to rapid loss of balance and early episode termination with minimal reward. In more complex scenarios, adversarial training methods such as those in (Gong et al., 2024) can be used to learn a reward-minimizing policy.

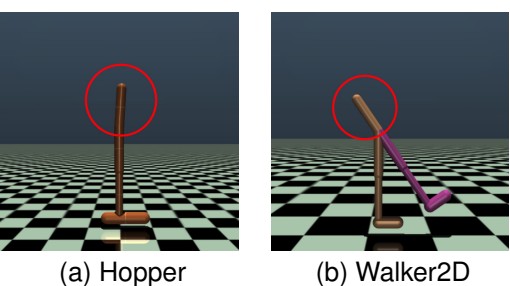

(a) Hopper    (b) Walker2D

Figure 2: The circled areas indicate where the triggers are inserted. The trigger is introduced by modifying the angle information corresponding to the circled points in the agent's observation.

The target backdoor policy facilitates backdoor implantation by assigning bad actions to triggered states, ensuring attack effectiveness. Simultaneously, it maintains backdoor stealthiness by preserving normal behavior in the absence of triggers.

### 4.2 REFORMULATION OF THE BACKDOOR ATTACK PROBLEM

In addition to poisoning the reward data in the agent's replay buffer, the attacker must minimize the distortion $\Delta$ introduced to the reward. This dual requirement naturally leads to a constrained optimization framework.

**Penalty Formulation.** The optimization objective is to minimize the data distortion $\Delta$ across all state-action pairs while ensuring the agent effectively learns the target policy $\pi^{\dagger}$. To achieve this, the induced $Q$-value function must satisfy the following constraints: for any $s \in \mathcal{S}$ and $a \in \mathcal{A}$,

$$Q(s,a) = r(s,a) + \Delta(s,a) + \gamma \sum_{s'} P(s'|s,a) Q(s', \pi^{\dagger}_{s'}),$$

and for any $s \in \mathcal{S}$ and $a \in \mathcal{A}$ where $a \neq \pi_s^\dagger$, the induced $Q$-value function is constrained by

$$Q(s, \pi_s^\dagger) \geq Q(s, a) + \epsilon, \tag{3}$$

where $\epsilon$, referred to as the poison intensity parameter, quantifies the advantage of $\pi_s^\dagger$ over other actions. The equality enforces adherence to the Bellman equation (1), while the inequality ensures the optimality of $\pi^\dagger$. A penalty method is applied to formulate the problem as follows:

$$\min_{\lambda, \theta} \quad \frac{1}{2} \sum_{s,a} (\Delta_{s,a}^\theta)^2 + \frac{\rho}{2} \sum_{s, a \neq \pi_s^\dagger} \Phi(\bar{Q}_{s,a}^\lambda + \epsilon - \bar{Q}_{s,\pi_s^\dagger}^\lambda)^2 \tag{4}$$

$$\text{s.t.} \quad \bar{Q}_{s,a}^\lambda = r(s,a) + \Delta_{s,a}^\theta + \gamma \sum_{s'} P(s'|s,a) \bar{Q}_{s',\pi_{s'}^\dagger}^\lambda, \ \forall s, a,$$

where $\Delta_{s,a}^\theta := \Delta(s, a; \theta)$ and $\bar{Q}_{s,a}^\lambda := \bar{Q}(s, a; \lambda)$ are parameterized as neural networks, with $\theta$ and $\lambda$ being their respective parameters. The term $\bar{Q}$ is an auxiliary variable maintained by the attacker, distinct from the agent's $Q$ function if any. The parameter $\rho$ represents the penalty magnitude, and $\Phi(x) := \mathbf{1}(x > 0)x$, whose square pertains to penalty for (3).

Although equality constraints in (3) can be handled as a multiple of penalty terms, the gradient of the resulting squared term with respect to $\bar{Q}$ requires two sampled transitions for an unbiased gradient estimator, a challenge known as the double sampling issue (Dai et al., 2018). To address such complications, we adopt a bi-level reformulation.

**Bi-level Reformulation.** The bi-level reformulation decomposes the problem into two hierarchical levels, where the upper-level problem updates $\bar{Q}$ variable to minimize objective function and penalty functions, and the lower-level one updates $\Delta$ to realize feasibility of the equality constraint. The bi-level optimization is as follows:

$$\min_{\lambda} \quad \frac{1}{2} \sum_{s,a} (\Delta_{s,a}^\theta)^2 + \frac{\rho}{2} \sum_{s, a \neq \pi_s^\dagger} (\Phi(\bar{Q}_{s,a}^\lambda + \epsilon - \bar{Q}_{s,\pi_s^\dagger}^\lambda))^2$$

$$\text{s.t.} \quad \theta \in \arg\min \left\{ \sum_{s,a} \frac{1}{2} \left( r(s,a) + \Delta_{s,a}^\theta + \gamma \sum_{s'} P(s'|s,a) \bar{Q}_{s',\pi_{s'}^\dagger}^\lambda - \bar{Q}_{s,a}^\lambda \right)^2 \right\}. \tag{5}$$

Since the lower-level problem admits a straightforward solution $\Delta_{s,a}^{\theta,*} = \bar{Q}_{s,a}^\lambda - r(s,a) - \gamma \sum_{s'} P(s'|s,a) \bar{Q}_{s',\pi_{s'}^\dagger}^\lambda$, the equivalence between (5) and (4) is clear. However, while the bi-level formulation avoids the double-sampling issue, it complicates gradient derivation due to the nested dependency of $\Delta_{s,a}^\theta$ on $\lambda$. This challenge can be addressed via the implicit function theorem (Hong et al., 2023; Liu et al., 2022; Ghadimi & Wang, 2018), which enables exact gradient computation for both levels by deriving the gradient coupling $\nabla_\lambda \Delta_{s,a}^{\theta,*}$.

### 4.3 UPDATE RULE

We leverage a single-loop algorithm to solve the reformulated be-level optimization problem. Due to the lack of access to transition probabilities, the attacker needs to compute stochastic gradients using sampled transitions, where $s' \sim P(\cdot|s,a)$. The gradient-descent update rule is summarized as follows:

$$\Delta_{s,a}^{\text{target},k} = \bar{Q}_{s,a}^{\lambda_k} - r(s,a) - \gamma \bar{Q}_{s',\pi_{s'}^\dagger}^{\lambda_k} \tag{6}$$

$$\bar{Q}_{s,a}^{\text{target},k} = \bar{Q}_{s,a}^{\lambda_k} - \left( \Delta_{s,a}^{\theta_k} + \rho_k \left[ \mathbf{1}(a \neq \pi_s^\dagger) \Phi(Q_{s,a}^{\lambda_k} + \epsilon - Q_{s,\pi_s^\dagger}^{\lambda_k}) \right. \right.$$

$$\left. \left. - \mathbf{1}(a = \pi_s^\dagger) \sum_{\tilde{a} \neq a} \Phi(Q_{s,\tilde{a}}^{\lambda_k} + \epsilon - Q_{s,a}^{\lambda_k}) \right] \right) \tag{7}$$

$$Q_{s',\pi_{s'}^\dagger}^{\text{target},k} = Q_{s',\pi_{s'}^\dagger}^{\lambda_k} + \gamma \Delta_{s,a}^{\theta_k}, \tag{8}$$

where $k$ is the iteration count. To satisfy the inequality constraints, the penalty coefficient $\rho_k$ should be gradually increased until it reaches a sufficiently large value. The update rule (6) is derived

---

**Algorithm 1** Backdoor Attack Algorithm via Bi-level Optimization

---

**Input**: Initial neural network parameters $\theta_0, \lambda_0$, poison intensity $\epsilon$, step sizes $\alpha, \beta$, penalty coefficients $\{\rho_k\}$, initial agent policy $\pi_0$

1: **for** training round $k = 1, 2, ...$ **do**
2:    **Data Collection**:
3:    Agent interacts with environment using $\pi_{k-1}$, stores transitions $\{\langle s_i, a_i, r_i, s_i' \rangle\}_{i \in I_k}$
4:    **Attacker: Reward Poisoning**:
5:    Compute $\Delta^{\text{target}}$ via (6)
6:    Update $\theta_k$:

$$\theta_{k+1} \leftarrow \theta_k - \alpha \nabla_\theta \frac{1}{2} \sum_{i \in I_k} [\Delta(s_i, a_i; \theta_k) - \Delta^{\text{target},k}_{s_i,a_i}]^2.$$

7:    Inject $\Delta(s_i, a_i; \theta_k)$ into rewards $\{r_i\}_{i \in I_k}$
8:    **Attacker: $Q$-value Poisoning**:
9:    Compute $\bar{Q}^{\text{target}}$ via (7) and (8)
10:   Update $\lambda_k$:

$$\lambda_{k+1} \leftarrow \lambda_k - \beta \nabla_\lambda \frac{1}{2} \sum_{i \in I_k} [\bar{Q}(s_i, a_i; \lambda_k) - \bar{Q}^{\text{target},k}_{s_i,a_i}]^2 + [\bar{Q}(s_i', a_i'; \lambda_k) - \bar{Q}^{\text{target},k}_{s_i',a_i'}]^2,$$

11:   **Agent Policy Update**:
12:   Agent updates policy $\pi_k$ using poisoned transitions $\{\langle s_i, a_i, \tilde{r}_i, s_i' \rangle\}_{i \in I_k}$
13: **end for**

---

from the optimality conditions of the lower-level problem, while the target values in (7) and (8) are updated by subtracting the stochastic gradient from the current values.

In the proposed reward poisoning framework, the poisoned reward $\bar{r}$ is adaptively calibrated using the poisoned Q-function $\bar{Q}$ to strategically influence the agent's behavior: when $\bar{Q}(s_t, a_t)$ for the target action $a_t$ is comparatively low (signifying a suboptimal estimated value), the reward $\bar{r}$ is increased to amplify the perceived desirability of $a_t$ and incentivize its selection. Conversely, if $\bar{Q}(s_t, a_t)$ is relatively high (indicating the agent already sufficiently values $a_t$), $\bar{r}$ is explicitly *decreased* toward the original reward $r$ to minimize unnecessary perturbation, thereby reducing detectability while maintaining attack efficacy. This dual adjustment ensures minimal reward manipulation: aggressive amplification occurs only when necessary to promote $a_t$, and conservative attenuation is applied when the agent's existing value estimates align with adversarial objectives.

Finally, the neural network parameters $\theta, \lambda$ are iteratively adjusted to online learn $\bar{Q}^{\text{target},k}_{s,a}$ and $\Delta^{\text{target},k}_{s,a}$ by minimizing the mean squared residue loss. The overall algorithm is summarized in Algorithm 1.

# 5 EXPERIMENTS

## 5.1 EXPERIMENTAL SETUP

**Tasks.** We conduct experiments on a classic control task(*CartPole* (Brockman et al., 2016)) and two robotic control tasks (*Hopper* and *Walker2D*) from MuJoCo (Todorov et al., 2012). The *Walker2D* environment features a larger observation and action space compared to *Hopper*, making RL training more challenging. We use these three environments to evaluate the performance of our backdoor algorithm across varying levels of complexity. We provide other details in Appendix A.

**Metrics.** In RL, we often use *cumulative return* to evaluate the performance of the RL agent. However, to measure the effectiveness of our backdoor attack algorithm, we need to consider more aspects of performance. The backdoor attack primarily affects the agent's performance: when the trigger is present, the poisoned agent should exhibit a significant performance drop; otherwise, its performance should closely match that of a normal agent. Therefore, it is essential to evaluate the relative change in the agent's performance.

Table 1: The attack results with different poison intensity parameters ($\epsilon$) in *Hopper* and *Walker2D*. We evaluate the effectiveness and stealthiness of our attack by assessing cumulative rewards under triggered scenarios(activated trigger) and normal scenarios (inactive trigger).

| Environments | Trigger | Normal | Poison Intensity Parameter ($\epsilon$) | | | | | |
|---|---|---|---|---|---|---|---|---|
| | | | 0.01 | 0.1 | 0.25 | 0.5 | 2.0 | 4.0 |
| *CartPole* | Activated | 464 | 442(**-4.74%**) | 400(**-13.79%**) | 362(**-21.98%**) | 345(**-25.65%**) | 304(**-34.48%**) | 136(**-70.69%**) |
| | Inactive | 471 | 463(**-1.70%**) | 458(**-2.76%**) | 462(**-1.91%**) | 460(**-2.33%**) | 465(**-1.27%**) | 468(**-0.63%**) |
| *Hopper* | Activated | 3449 | 963(**-72.08%**) | 1091(**-68.35%**) | 517(**-85.01%**) | 610(**-82.31%**) | 497(**-85.57%**) | 741(**-78.50%**) |
| | Inactive | 3486 | 2696(**-22.64%**) | 2669(**-23.42%**) | 3251(**-6.72%**) | 3410(**-2.18%**) | 3272(**-6.14%**) | 2811(**-19.36%**) |
| *Walker2D* | Activated | 3350 | 1213(**-63.78%**) | 1307(**-60.97%**) | 962(**-71.27%**) | 507(**-84.87%**) | 544(**-83.76%**) | 363(**-89.16%**) |
| | Inactive | 3541 | 3221(**-9.02%**) | 3407(**-3.78%**) | 3378(**-4.59%**) | 2635(**-25.57%**) | 2813(**-20.56%**) | 2694(**-23.91%**) |

- *Backdoor Effectiveness.* We use this metric to measure the impact of activated triggers on the performance of the backdoored agent. However, we must also account for how much the trigger itself affects the normal agent to accurately reflect the true impact of the backdoor attack. Let $R_{T_n}$ denote the cumulative return of the normal policy agent with activated triggers, and $R_{T_b}$ represent the cumulative return of the backdoored policy agent with activated triggers. We define the performance decrease as $100\% \times \frac{R_{T_n} - R_{T_b}}{R_{T_n}}$. The larger the gap, the more effective the backdoor attack is.

- *Backdoor Stealthiness.* We use this metric to evaluate the stealthiness of backdoor attacks. Specifically, we aim to measure the gap between the performance of the backdoored strategy and the normal strategy when no triggers are activated. Let $R_{N_n}$ denote the cumulative return of the normal policy agent without triggers (under normal scenarios), and $R_{N_b}$ represents the cumulative return of the backdoored policy agent under normal scenarios. We define the performance gap as $100\% \times \frac{R_{N_n} - R_{N_b}}{R_{N_n}}$. A smaller gap indicates higher stealthiness of the backdoor attack.

- *Policy Distance.* The RL algorithm ultimately converges to a policy, and we aim to evaluate the impact of the backdoor attack by measuring the deviation in the learned policy. Specifically, we uniformly sample trigger states and normal states. For the trigger states, we compute the difference between the agent policy action and the predefined bad action. For the normal states, we compute the difference between the agent's policy action and the action derived from the normal policy. These differences represent the gap between the trained agent's policy and the target backdoor policy, as well as the gap between the trained agent's policy and the normal policy. Our goal is to minimize both gaps during training, with action distances quantified using the $\ell_2$-norm.

**RL Training and Testing Setup.** We train the *CartPole* task using the deep Q-learning algorithm, and employ Proximal Policy Optimization (PPO) (Schulman et al., 2017) for the *Hopper* and *Walker2D* tasks. During training, the attacker accesses the rewards data used for training and modifies it according to our attack algorithm. The training procedure stops when a certain test reward or a maximum number of iterations is reached. During the test phase, when the time step reaches a certain upper limit or the agent's status becomes unhealthy (e.g., when the agent falls to the ground and cannot move), the test will be stopped. All experiments are repeated 5 times to ensure statistical reliability.

**Backdoor Attack Setup.** For *CartPole*, the bad action $a_{\text{bad}}$ is defined as a fixed action that pushes the cart to the right whenever the trigger is activated. This action rapidly destabilizes the pole, causing it to deviate beyond the allowed angle threshold, thus terminating the episode prematurely and resulting in a significantly reduced reward.

For MuJoCo tasks, the bad action $a_{\text{bad}}$ is defined as $[1, -1, -1]$ for *Hopper* and $[-1, -1, -1, -1, -1, -1]$ for *Walker2D*. These bad actions are designed to cause the agent to fall immediately after the trigger is activated, achieving the attack's objective. We configure the *Hopper* and *Walker2D* environments with dispersions of 8 for each action dimension to compute (7).

The penalty coefficient $\rho$ is set to 20. The learning rates for the poisoned reward network and the $Q$-value network are set to $10^{-4}$ and $10^{-5}$, respectively.

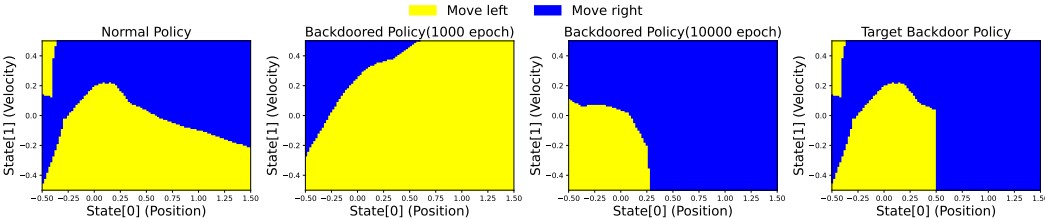

Figure 3: Policy distance results during training in *CartPole*. As training progresses, the policy distribution of the backdoored agent gradually approaches that of the target backdoor policy.

## 5.2 EXPERIMENTAL RESULTS

In this section, we evaluate the effectiveness and stealthiness of our backdoor method. First, we analyze the impact of the backdoor algorithm on the RL agent from a macro perspective, focusing on its performance during the deployment phase. Next, we investigate how our attack algorithm influences the agent's policy iteration during the training process. Finally, we examine the impact of the poison intensity parameter on the performance of the backdoor attack.

**Backdoor Effectiveness.** Table 1 presents the overall results of our backdoor attack method in the *CartPole*, *Hopper*, and *Walker2D* environments. The experiments show that the performance of the normal policy remains nearly identical regardless of the presence of triggers, indicating that the triggers themselves have minimal impact on the agent.

After the backdoor attack, the agent's performance degrades significantly, e.g., by $85.01\%$ in the *Hopper* task and by $71.27\%$ in the *Walker2D* task, under the poison intensity parameter $\epsilon = 0.25$. Moreover, regardless of the parameter settings, the attack consistently achieves performance degradations of at least $68.35\%$ and $60.97\%$ in the *Hopper* and *Walker2D* tasks, respectively.

**Backdoor Stealthiness.** Table 1 demonstrates that our backdoor attack algorithm is highly stealthy, as the backdoored policy behaves similarly to the normal policy under normal scenarios. The performance drop is merely $6.72\%$ and $4.59\%$ for the *Hopper* and *Walker2D* tasks, respectively. Furthermore, across all parameter settings, the performance degradation never exceeds $23.42\%$ and $23.91\%$ for the two tasks. This is attributed to the target backdoor policy designed in Section 4.1, which ensures that the agent performs normally and maintains its performance in the absence of triggers.

**Policy Distance Results.** Figure 3 illustrates the policy distributions for the *CartPole* environment under normal, poisoned ($\epsilon = 4$), and target backdoor policies. Given the discrete action space ("move left" and "move right"), we analyze the policy distance by examining action distributions across cart positions ranging from $-0.5$ to $1.5$ and velocities from $-0.5$ to $0.5$, with other states fixed at zero. The results for other tasks are in the Appendix B.

The target backdoor policy sharply changes near position $x = 0.5$, consistently forcing the agent to move right once exceeded, causing rapid episode termination. For positions below $0.5$, the target backdoor policy matches the normal policy to maintain stealthiness.

As training progresses (from 1000 to 10000 epochs), the poisoned policy gradually approaches the target policy. While a gap remains due to difficulties in learning abrupt policy transitions, the poisoned policy effectively achieves the intended behavior. For positions greater than $0.5$, the poisoned policy consistently selects the "move right" action, confirming the attack's effectiveness. Below $0.5$, it aligns closely with the normal policy, preserving stealthiness when the trigger is inactive.

**Impact of the Poison Intensity Parameters ($\epsilon$).** Now we explore the effect of the poison intensity $\epsilon$ on the backdoor attack. According to the deployment phase results in Table 1, no matter how the parameters are set, the effectiveness and stealthiness of backdoor attacks are basically satisfied. When the parameters are relatively small (e.g., $\epsilon = 0.01$), the effectiveness of the backdoor is limited. Conversely, when the parameters are large, (e.g., $\epsilon = 4$), the backdoor achieves higher effectiveness but sacrifices some stealthiness. This is because larger parameters introduce more data manipulations, which can lead to increased instability during training.

We observe that the optimal poison intensity $\epsilon$ varies across RL tasks. Smaller values (e.g., $\epsilon < 1.0$) already produce noticeable backdoor effects in *Hopper* and *Walker2D* (Table 1), but show limited impact in *CartPole*, achieving only $4.74\%$ effectiveness at $\epsilon = 0.01$. This is due to differences in the $Q$-value ranges across environments. However, overly large $\epsilon$ values compromise stealthiness.

### 5.3 ATTACK INTENSITY

To evaluate the efficacy of the proposed method, we conduct a comparative analysis against several baseline poisoning attacks within the *CartPole* environment. The baselines include:

1. Neighbourhood-based attacker (Xu & Singh, 2023): Penalizes non-target actions in targeted states with a fixed value.

2. Min-max attacker: Assign maximal rewards for target actions and minimal rewards for all other actions in targeted states, a technique partially employed in prior work (Kiourti et al., 2020).

3. Random attacker: Modifies the reward by adding a bounded, uniformly sampled value for target actions and a random penalty for other actions.

The experimental results, summarized in Table 2, demonstrate that all evaluated methods successfully install a backdoor. Specifically, the poisoned agent exhibits nominal performance comparable to a benign agent (achieving the maximum reward of 475) in the absence of the trigger. However, upon activation of the trigger, the agent's performance degrades catastrophically. The random attacker induces the largest reduction in cumulative rewards (62) but also deviates markedly from nominal behavior, yielding only 304 rewards in the absence of the trigger and thus becoming highly susceptible to detection. By contrast, the proposed method maintains near-nominal behavior when the trigger is inactive (468/500) while achieving a greater reduction in rewards than both the neighborhood-based and min–max attackers.

To quantify the stealthiness of the attacks, we measure the perturbation intensity, defined as the L2 norm of the deviation from the original reward function: $\sum_{s,a} \|\bar{r}_{s,a} - r_{s,a}\|_2^2$ . This metric was evaluated for both trigger-specific state-action pairs and global pairs sampled uniformly from the entire state space.

Table 2: The poisoned agent's performance and poisoning intensity in the *CartPole* environment. The poisoning intensity is computed by uniformly sampling over triggered/universal states and summing up the poisoned rewards.

| Method | Reward sum (Trigger Status) | | Intensity | |
|---|---|---|---|---|
| | Activated | Inactive | Global | Triggered |
| Neighbourhood | 348 | **475** | 4.004 | 2.10 |
| Minmax | 237 | 446 | 5.00 | 5.00 |
| Random | **62** | 304 | 9.99 | 9.98 |
| **Proposed** | 136 | 468 | **2.48** | **0.87** |

Our analysis reveals that the proposed method achieves a lower perturbation intensity across both distributions. This result indicates that our approach can induce the targeted malicious behavior with minimal modification to the original reward function, demonstrating superior stealth and efficiency. This advantage arises because our method leverages the underlying MDP dynamics to distribute subtle alterations across many non-triggered states, which collectively influence behavior under the trigger condition. In contrast, baseline methods must concentrate larger, more conspicuous perturbations exclusively on the triggered states, rendering their manipulations more readily identifiable.

## 6 CONCLUSION

In this paper, we revisit the RL backdoor attack problem by formulating it through an optimization framework. Our approach minimizes data distortions while ensuring both the effectiveness and stealthiness of the implanted backdoor. We design a target backdoor policy and propose a novel iterative optimization algorithm using a penalty-based approach and bi-level reformulation. Our experiments demonstrate that the attack introduces minimal disruption to the agent's normal behavior while significantly degrading its performance when triggered. Our work offers both theoretical insights and practical methods for analyzing and evaluating backdoor attacks in RL.

REPRODUCIBILITY STATEMENT

The implementation code has been included in the supplementary materials. Comprehensive details regarding the experimental setup, datasets, baselines, hyperparameters, training procedure, and attack setup are provided in Section 5.1 and Appendix A. We commit to releasing the full implementation upon publication of this work.

ETHICS STATEMENT

This paper investigates a stealthy backdoor attack against RL agents, with the goal of understanding potential vulnerabilities in deployed decision-making systems. Our intention is not to promote misuse, but to provide insight into how such attacks can arise and inform the design of future defenses. We emphasize that the proposed attack is studied solely in simulated environments (e.g., *CartPole*, *Hopper*, and *Walker2D*) using publicly available benchmarks and models. No real-world systems or proprietary data are involved. While such attacks could be exploited maliciously, our research aims to raise awareness and motivate the development of robust defense mechanisms in safety-critical RL applications.

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

## A    ADDITIONAL EXPERIMENTAL SETUP

**Task Details.** We briefly describe the environments used in our experiments, including their physical setup, observation space, action space, and control objectives:

- *CartPole:* In *CartPole*, a pole is hinged to a movable cart, constrained to one-dimensional horizontal motion along a frictionless track. The objective is to keep the pole balanced vertically by applying discrete horizontal forces to the cart. The observation is a 4-dimensional vector representing cart position, cart velocity, pole angle, and pole angular velocity, while the action space consists of two discrete actions: pushing the cart left or right.

- *Hopper:* In *Hopper*, the robot is a two-dimensional, single-legged entity comprising four principal components: the torso at the top, the thigh in the center, the leg at the lower end, and a single foot on which the entire body rests. The objective is to maneuver the robot forward (to the right) by exerting torques on the three hinges that interconnect these four body segments. The observation in *Hopper* is an 11-dimensional vector, while the action is a 3-dimensional vector.

- *Walker2D: Walker2D* introduces a greater number of independent state and control variables to more accurately emulate real-world scenarios. The robot in *Walker2D* is also two-dimensional but features a bipedal design with four main components: a single torso at the top from which the two legs diverge, a pair of thighs situated below the torso, a pair of legs below the thighs, and two feet attached to the legs that support the entire structure. The objective is to coordinate the movements of both sets of feet, legs, and thighs to progress forward by applying torques to the six hinges that connect these body parts. In *Walker2D*, the observation is represented as a 17-dimensional vector, and the action is represented as a 6-dimensional vector.

**Trigger Setup.** To better distinguish triggers from normal states, we select states that are rarely encountered by policies trained under normal conditions as triggers. This design enhances the stealthiness of the backdoor attack.

For *CartPole*, we define the trigger based on the cart's horizontal position. Specifically, the trigger is activated when the cart's position exceeds $0.5$ to the right; conversely, when the position remains below $0.5$, the agent is expected to maintain normal behavior. During the deployment phase, the trigger condition is enforced by setting the cart's position explicitly to the predefined trigger position at the initial time step.

For *Hopper* and *Walker2D*, we define the trigger based on the head's bending angle. Specifically, for *Hopper*, the trigger is activated when the head angle is between $3°$ and $4°$; for *Walker2D*, the trigger is activated when the head leans back more than $34°$. During the deployment phase, the trigger is activated at a specific time step by modifying the agent's head angle to the predefined trigger angles. The illustration of the triggers for *Hopper* and *Walker2D* is shown in Figure 2.

## B    ADDITIONAL EXPERIMENTAL RESULTS

Figure 4 illustrates the change in policy distance for *Hopper* and *Walker2D* during training under different parameter settings. The distances exhibit fluctuations due to the inherent instability of RL training. Overall, the attacker successfully guides the agent to converge toward the target backdoor policy during the training process. Specifically, the distances between the agent's policy and the target backdoor policy, as well as between the agent's policy and the normal policy, decrease over time, despite some fluctuations. For example, in the *Hopper* task, when $\epsilon = 0.25, 2, 4$, the backdoored agent's behavior under normal scenarios gradually aligns with the normal policy; when $\epsilon = 0.25, 2$, the agent's behavior under triggered scenarios also progressively approaches the target backdoor policy. These results on policy distance explain the effectiveness and stealthiness of our backdoor attacks.

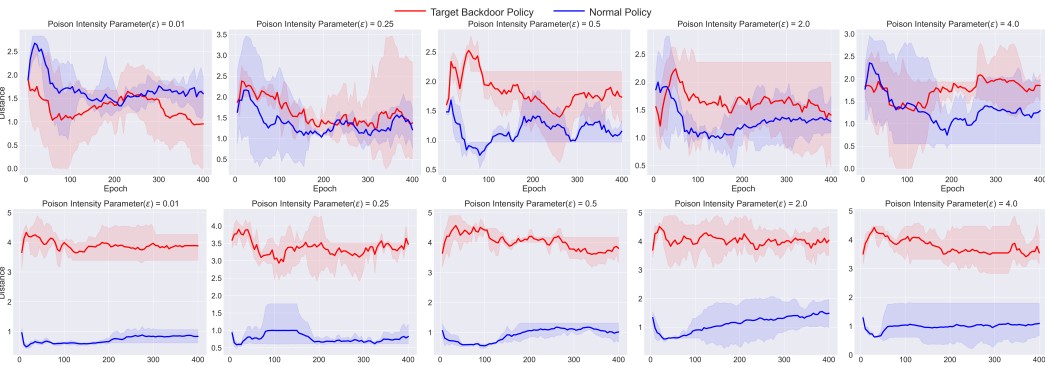

Figure 4: Policy distance results during training with different poison intensity parameters ($\epsilon$) in *Hopper* and *Walker2D*.

## C   LIMITATIONS & FUTURE WORK

1) Our attack assumes access to the training buffer — an assumption common in prior online RL backdoor work, but not always realistic. Future work could consider weaker threat models, such as partial access to offline data. 2) This work focuses on attack design; studying defense mechanisms, like runtime anomaly detection or policy consistency checks, remains an important direction. 3) The induced behaviors are mostly unstructured (e.g., failure to balance), which are relatively easy in RL. Exploring structured backdoor goals, such as task redirection, would better showcase an attacker's full potential.

## D   USE OF LARGE LANGUAGE MODELS(LLMS)

We used LLMs (e.g., ChatGPT) only for editing and language refinement. No LLMs were involved in research ideation, technical development, or experimental design.

