# OpenReview forum: "Stealthy Backdoor Attack in Reinforcement Learning via Bi-level Optimization"
_ICLR.cc/2026/Conference — Submitted to ICLR 2026_

### Official Review · Reviewer_dSW9 · 2025-10-28

**Soundness:** 2
**Presentation:** 2
**Contribution:** 1
**Rating:** 2
**Confidence:** 5

**Summary:**

The paper develops a new backdoor attack method against deep reinforcement learning aimed at minimizing reward perturbation magnitudes needed for a successful attack. The method learns a q-function to determine the minimum reward perturbation necessary to induce a successful attack. Evaluations are performed across some standard, continuous action space domains and the method is compared against some prior literature.

**Strengths:**

* The paper's goals of minimizing reward perturbations along with maximizing agent benign performance is well motivated.

* The method appears reasonably easy to implement and achieves respectable results.

* The proposed approach of bi-level optimization is reasonable.

**Weaknesses:**

### Significant Omissions in Literature Review

The main weakness of this paper is it highly outdated literature review and comparisons against existing attacks. The paper only compares its methods against methods from up until 2023, while many, better performing and more principled methods have been developed since then. Specifically, and most worryingly, are omissions of SleeperNets [1] and Q-Incept [2] which represent the current state of the art for backdoor attacks against DRL, being the only methods with provable guarantees of attack success in general MDPs. Additionally, methods like Q-Incept were developed to solve the very problem proposed in this paper, i.e. minimizing the scale of perturbations applied to the agent's rewards. Furthermore, BadRL [3] was also proposed as a method to improve attack stealth. Therefore, given the existence of these works, I find the contributions of this paper to be very minimal.

The generalized nature of these methods along with their principled approaches additionally makes many of the author's claims untrue, e.g. "research on backdoor attacks in RL remains limited, often focusing on specific tasks, or heuristic methods".

One concession I will make is that neither SleeperNets nor Q-Incept have been evaluated against continuous action space domains, to the best of my knowledge, so some non-trivial extensions maybe required. That being said, the same remains true for methods like TrojDRL, which was also developed for discrete action space environments, so I believe comparing against at least one of these methods (preferably Q-Incept) is a fair request.

[1] Sleepernets: Universal Backdoor Poisoning, Ethan Rathbun, Christopher Amato, and Alina Oprea.
Attacks Against Reinforcement Learning Agents. In Advances in Neural Information Processing Systems, 2024.

[2] Adversarial Inception Backdoor Attacks against Reinforcement Learning,
 Ethan Rathbun, Alina Oprea, and Christopher Amato. In International Conference on Machine Learning, 2025.

[3] "Badrl: Sparse targeted backdoor attack against reinforcement learning.", Cui, Jing, et al.
Proceedings of the AAAI Conference on Artificial Intelligence. Vol. 38. No. 10. 2024.

### Experimental Results

1.) The proposed method is evaluated against too few environments and does not cover a diverse enough range of environments. I suggest the authors consider further evaluations in discrete action space domains (e.g. Atari) along with a few more MuJoCo tasks e.g. Half cheetah and or Ant.

2.) As previously mentioned, the paper includes insufficient comparisons against prior work. Only comparing the proposed method against older attacks which have since been superseded and only performing this comparison in one environment.

3.) In all evaluated environments the triggered state space has non-empty intersection with the natural state space of the environment. This makes evaluating the attacks more difficult as the agent now has to learn a benign policy that accounts for poor performance in triggered states. For instance, in cart-pole an optimal agent should learn to just avoid entering a horizontal position above 0.5 or else it either a.) swiftly moves right and fails the task or b.) ignores the target action and receives a large penalty. This may be why we see such a large dip in "inactive" performance on cart pole.

This is an inherent flaw of the authors' chosen environments as the agents only receive self-referential, proprioceptive observations which cover the entire feasible state/observation space. For instance, in MuJoCo tasks the agent's observations are in the range $\[- \infty, \infty\]$, meaning there are no "triggered states" that are distinct from all benign states. This gets away from the initial goal/formulation of backdoor attacks in which the trigger is something an adversary can manually trigger at test time. The authors seem to align themselves with this problem formulation in section 3.2 and the beginning of section 4, where triggered states are represented by $\tilde{s} = s + \delta$ indicating that the trigger $\delta$ "adds" something to the benign state $s$.

For this reason many prior attacks studying backdoor attacks in DRL, including TrojDRL, BadRL, SleeperNets, and Q-Incept have focused on environments where the agent receives external observations of the environment, e.g. pixel images from Atari games. If the authors wish to continue using MuJoCo tasks I suggest they either evaluate on image versions of MuJoCo tasks or include a "binary flag" trigger, appending an extra [0,1] binary flag to the end of the agent's state to indicate triggered vs benign states. While this does extend the state space, I believe it is a valid way to abstractly represent the trigger for evaluation purposes while remaining faithful to the general formulation of backdoor attacks as having external and distinct triggers.

If the authors wish to learn a different objective without distinct triggers they should make this clear and reformulate their problem statement and attack objectives accordingly.

3.) It would be useful if the authors included an attack success rate metric in their evaluations to make for easier comparison against existing works. For continuous action space environments this can be evaluated by thresholding the l-infinity distance between the chosen action adn the target action.

### Method

1.) The authors drop their usage of $\tilde{s}$ during the development of their method which makes the existence and role of triggered states much less clear. As it is currently written, the attack seems much more similar to policy replacement attacks like those explored in [4]. Note that I am not saying that the author's work represents a policy replacement attack, rather I recommend they adjust their notation to make the role of the backdoor trigger more clear when developing their method.

2.) The way the attack is formulated makes it seem that the attacker must learn a benign policy $\pi_n$ in the environment before the attack occurs. I don't believe this is true for the author's proposed method so I suggest making this more clear.

[4] Understanding the Limits of Poisoning Attacks in Episodic Reinforcement Learning. Rangi, Anshuka & Xu, Haifeng & Tran-Thanh, Long & Franceschetti, Massimo. (2022). 3369-3375. 10.24963/ijcai.2022/468.

**Questions:**

* Is the attacker's goal to minimize return or to maximize the probability of the target action given the trigger?

* Does the attacker need to learn a benign policy $\pi_n$ before executing the attack?

---

### Official Review · Reviewer_KQuB · 2025-10-30

**Soundness:** 2
**Presentation:** 2
**Contribution:** 2
**Rating:** 2
**Confidence:** 4

**Summary:**

This paper proposes a new backdoor attack against RL agents. The main idea is to poison the rewards of data samples in the training agent's replay buffer. The attack is formulated as a bilevel optimization formulation to manipulate the agent's behavior towards a target policy, while
ensuring that the learned Q function under the poisoned rewards and the target policy satisfies the Bellman equation to remain stealthy.

**Strengths:**

Unlike previous heuristics-based attacks, the proposed backdoor attack is grounded in a bilevel optimization formulation with meaningful objective and constraints.

**Weaknesses:**

1. The main weakness of the paper is that the threat model is unjustified. The paper assumes that the attacker can directly poison an agent's replay buffer. Further, it can modify the rewards of all transitions in the replay buffer. Are there any real-world scenarios where these can happen? Why is such an attack considered stealthy? In traditional backdoor attacks, the attacker can typically poison only a small fraction of the offline data. Here, the attacker can poison all the training data in real time. In addition, how can the attacker update the poisoned reward and Q networks in real time using data from the replay buffer while the agent is being trained? This seems a much stronger assumption than knowing the agent's training algorithm.
2. The stealthiness constraint is also not well justified. Traditionally, stealthiness is achieved by constraining the amount of perturbation introduced. Why can maintaining the Bellman equation of the surrogate Q function under the target policy help ensure stealthiness, which is more like a constraint for ensuring effectiveness?
3. One limitation of the proposed attack is that the trigger for an environment is manually chosen and is activated in a prespecified subset of states, which requires a significant amount of domain knowledge. Further, this approach does not generalize to environments with image input.
4. The definition of black-box attacks is confusing. The paper claims that the attacker is model- and environment-agnostic. However, Section 4.1 says that "the attacker begins by following the standard RL training procedure to obtain a normal policy $\pi_n$." How can the attacker obtain the normal policy without access to the environment? The paper mentions that the attacker can also learn a reward-minimizing policy. How can this be achieved without access to both the environment and the agent's policy (unless the attacker can attack at every time step)? An attack cannot be considered black-box if it has access to both the environment and the agent's policy.

**Questions:**

1. The paper assumes that the attacker can directly poison an agent's replay buffer while the agent is being trained. Further, it can modify the rewards of all transitions in the replay buffer. Why is this considered a realitic and stealthy attack?
2. How can the agent train a normal policy and specify a meaningful trigger without access to the environment?

---

### Official Review · Reviewer_Dan8 · 2025-10-30

**Soundness:** 2
**Presentation:** 3
**Contribution:** 2
**Rating:** 4
**Confidence:** 3

**Summary:**

This paper presents a novel bi-level formulation of RL backdoor attack that achieves supreme backdoor attack performance and achieves the best stealthiness compared with previous methods. The experiment results on Mojuco environments empirically show that the proposed attack is both strong and stealthy.

**Strengths:**

1. The adaptive reward manipulation during the training is novel and successfully injects the backdoor attack to the trained policy and remains stealthy.

2. The bilevel optimization is well-formulated, and the intuition is well-explained. The update rule section clearly describes how the a single-loop algorithm can solve the bilevel optimization problem.

**Weaknesses:**

1. Typo line 257: be-level -> bi-level

2. Although the specific formulation of the bi-level optimization in this work's setting is new, there are many previous works in traditional backdoor attack that used similar bi-level optimization methods, such as [1][2]. I think the author should include a related work section discussing the current bi-level formulation.

[1] Hayase, J. & Oh, S. Few-shot Backdoor Attacks via Neural Tangent Kernels.
[2] Sun, W., Zhang, X., Lu, H., Chen, Y., Wang, T., Chen, J. & Lin, L. Backdoor Contrastive Learning via Bi-Level Trigger Optimization.

3. Experiment environments are limited. It will strengthen the work's soundness if the author can show that the proposed method can be applied and work in more complicated environments.

**Questions:**

1. What is the backdoor training overhead of the proposed method compared with other methods?

2. Could the proposed method scale to more complicated environments, such as image based RL environment?

3. Are there any early thoughts on how to mitigate the proposed defense? High-level ideas are good to me.

---

### Official Review · Reviewer_s2sF · 2025-10-31

**Soundness:** 2
**Presentation:** 3
**Contribution:** 2
**Rating:** 4
**Confidence:** 4

**Summary:**

This paper proposes a novel and principled framework for conducting stealthy backdoor attacks against Reinforcement Learning (RL) agents. The core contribution is the formalization of the backdoor attack as a bi-level optimization problem.

The attacker's goal is twofold: Effectiveness: The agent's performance should degrade significantly when a specific trigger is applied to its observed state. Stealthiness: The agent should behave normally in all non-triggered states , and the data poisoning itself should be minimally detectable by minimizing reward distortions.

The authors propose a single-loop iterative algorithm to solve this problem in a black-box setting (i.e., without knowledge of the agent's algorithm or environment dynamics). Experiments on CartPole, Hopper, and Walker2D show the attack is highly effective (e.g., up to 82.31% performance drop in Hopper) while remaining stealthy (e.g., only a 2.18% performance drop in normal Hopper scenarios).

**Strengths:**

Principled Optimization Framework: The primary strength of this paper lies in shifting RL backdoor attacks from heuristic approaches (e.g., simply assigning a large negative reward) to a formal optimization framework. This represents a significant contribution, offering a structured and analyzable method for understanding attack capabilities. Moreover, the authors incorporate the Bellman equation into the optimization process, which is both novel and conceptually sound.

**Weaknesses:**

Powerful Threat Model Assumption: The primary weakness is the power granted to the attacker. The paper assumes the attacker has continuous, online \textbf{access} to the agent's replay buffer and can inject custom-computed poisoned data at every training round. This attacker is not just a one-time data poisoner; it's an active, adaptive agent learning alongside the victim. This is a very strong assumption, which the authors rightly note in Appendix C. The "black-box" claim  is true for the agent's model, but it's a "white-box" attack on the training process.

Unstructured Attack Goals: The "bad actions" ($a_{bad}$) are pre-defined and simple, such as "move right" or "apply this fixed-torque vector". The goal is simply to make the agent fail. This is an acknowledged limitation. A more compelling (and dangerous) attack would involve a structured goal, such as forcing the agent to move to a different target location. The framework could likely support this, but it is not demonstrated.

**Questions:**

Please see the weakness

---

### Meta-Review · Area_Chair_eFX7 · 2025-12-20

**Summary:**

The reviewers pointed out several weaknesses and shared common concerns, including (a) an unrealistic assumption about the attacker’s power in the studied threat model; (b) experimental evaluation being limited to a few environments; (c) limited discussion and comparison with recent methods in the literature.

**Reviewer Concerns:**

There was no rebuttal for this paper, and I believe the reviewers’ concerns raised in the original reviews are still outstanding.

**Reviewer Scores:**

There was no rebuttal for this paper, and I don’t think the reviewers would have changed their scores without a rebuttal.

---

### Decision · Program_Chairs · 2026-01-26

Reject